# DisiMiR: Predicting Pathogenic miRNAs Using Network Influence and miRNA Conservation

**DOI:** 10.3390/ncrna8040045

**Published:** 2022-06-23

**Authors:** Kevin R. Wang, Michael J. McGeachie

**Affiliations:** 1Roxbury Latin, Boston, MA 02132, USA; 2Channing Division of Network Medicine, Brigham and Women’s Hospital and Harvard Medical School, Boston, MA 02115, USA; remmg@channing.harvard.edu

**Keywords:** miRNAs, computational biology, machine learning

## Abstract

MiRNAs have been shown to play a powerful regulatory role in the progression of serious diseases, including cancer, Alzheimer’s, and others, raising the possibility of new miRNA-based therapies for these conditions. Current experimental methods, such as differential expression analysis, can discover disease-associated miRNAs, yet many of these miRNAs play no functional role in disease progression. Interventional experiments used to discover disease causal miRNAs can be time consuming and costly. We present DisiMiR: a novel computational method that predicts pathogenic miRNAs by inferring biological characteristics of pathogenicity, including network influence and evolutionary conservation. DisiMiR separates disease causal miRNAs from merely disease-associated miRNAs, and was accurate in four diseases: breast cancer (0.826 AUC), Alzheimer’s (0.794 AUC), gastric cancer (0.853 AUC), and hepatocellular cancer (0.957 AUC). Additionally, DisiMiR can generate hypotheses effectively: 78.4% of its false positives that are mentioned in the literature have been confirmed to be causal through recently published research. In this work, we show that DisiMiR is a powerful tool that can be used to efficiently and flexibly to predict pathogenic miRNAs in an expression dataset, for the further elucidation of disease mechanisms, and the potential identification of novel drug targets.

## 1. Introduction

MicroRNAs (miRNAs) are small non-coding RNAs that regulate major cellular functions through action on multiple genes, and miRNA dysregulation has been heavily linked to many diseases, including Alzheimer’s disease, heart failure, and many types of cancer. Up to 60% of the proteome is estimated to be regulated by miRNAs, and these post-transcriptional regulators are differentially expressed in almost every disease studied to date [1]. Due to the regulation of disease-causing proteins, variation in miRNA expression can modify or ameliorate the disease state, thus forming attractive drug targets. Some proteins are completely undruggable, but they could be targeted through their miRNA regulators, enabling the treatment of diseases that, up to this point, might seem impossible to cure [2]. MiRNA-based drugs for cancers have the potential to target several genes in the same oncogenic pathway, thereby maintaining the drug’s efficacy even if multiple target genes develop mutations, thereby decreasing the development of acquired resistance [3]. In pursuit of pathogenic miRNAs, differential expression analysis has been an invaluable tool in narrowing the search. However, disease-associated miRNAs are not always pathogenic [4], since a miRNA differentially expressed by disease state can have statistical significance but not functional significance. Indeed, it can be hard to determine if dysregulation of a particular miRNA is the cause or the result of an underlying disease. Discovering disease causal miRNAs requires interventional experiments that can be time-consuming and costly, potentially taking many months and many thousands of dollars to execute [5]. Thus, computational methods to predict disease causal miRNAs are very attractive, although few methods exist.

DisiMiR is based on the hypothesis that pathogenic miRNAs have a set of biological characteristics that set them apart, at least on average, from other miRNAs. One such characteristic is that disease causal miRNAs are highly influential in the miRNA–miRNA interaction network, in which they regulate other miRNAs indirectly through the regulation of genes. Evidence of this characteristic is the fact that there are several miRNAs that are differentially expressed and functionally implicated in virtually all cancers, such as miR-21, meaning that these miRNAs are so influential that they influence several diseases [6]. MiRNAs that are highly influential have been suggested to cause signaling cascades, which, when dysregulated, can lead to the disruption of homeostasis and the progression of disease [7]. Highly influential miRNAs are also likely to regulate many mRNA targets, with the number of direct gene targets also contributing to the disease-causing potential of miRNAs. Additionally, many pathogenic miRNAs will be differentially expressed. Although being differentially expressed does not guarantee pathogenicity, it does increase the probability of it [4]. A final characteristic of pathogenic miRNAs is that they are more likely to be biologically conserved across species. Many disease causal miRNAs regulate key biological functions and, by virtue of their importance, are more likely to be conserved [8].

DisiMiR integrates these characteristics into a machine learning pipeline that works at the disease-specific level. DisiMiR uses miRNA conservation data, predicted target information, disease context miRNA expression data, and miRNA–disease causal associations to train a model to predict pathogenic miRNAs. At the disease-specific level, researchers can train DisiMiR on a single dataset to generate potential hypotheses for future investigation. DisiMiR’s implementation is available online at https://github.com/Wanff/DisiMiR (accessed on 6 June 2022).

Other methods used to predict pathogenic miRNAs computationally work strictly at the pan-disease level. miRfluence is a computational method that determines influential miRNAs by inferring miRNA influence in a cross-disease miRNA–miRNA interaction network that is able to identify influential miRNAs in multiple diseases [7]. According to our knowledge, miRfluence was the first method to predict disease causal miRNAs in a disease-agnostic framework. However, since there was no established database of disease causal miRNA associations at the time, like the Human MicroRNA Disease Database (HMDD) 3.2 Causal Database [9], they confirmed their results through corroboration with other computational methods, such as miRsig and TAM [10,11]. Building on recent data advances, MiRNA–Disease Causal Association Predictor (MDCAP) is a label propagation algorithm that utilizes miRNA and disease similarity matrices to predict potential disease causal associations [12]. Like other miRNA–disease association prediction algorithms, MDCAP uses clustering of miRNAs based on their associations to similar diseases to predict further miRNA–disease associations [13]. However, a benchmarking study of 36 miRNA–disease association prediction algorithms showed that none of the algorithms were able to differentiate between causal and non-causal miRNAs, with the highest AUC being 0.538. The authors of the study suggested that additional biological information is necessary for the prediction of disease causal miRNAs [14]. One great strength of MDCAP was the development of HMDD, a manual compilation of disease causal miRNAs [9]. MDCAP can effectively differentiate between the causal and non-causal miRNAs in the HMDD database; however, like other disease-association prediction algorithms, it is restricted by the size and quality of its training dataset, and thus struggles to generalize to diseases less characterized in the literature. The authors of MDCAP have also recently created an improved method, LE-MDCAP, that uses Levenshtein distance to measure miRNA sequence, function and expression similarities, and significantly outperforms the previous MDCAP model in differentiating between disease causal vs. non-causal miRNAs and disease causal vs. disease-associated miRNAs [15]. All three methods, miRfluence, MDCAP, and LE-MDCAP, have discovered important characteristics about pathogenic miRNAs: miRfluence showed that miRNA pathogenicity was linked to miRNA network influence; MDCAP discovered that miRNA pathogenicity was associated with cross-species conservation; and LE-MDCAP demonstrated the efficacy of the Levenshtein distance as a measure of miRNA similarity [7,12,15]. We build on these insights in our work with DisiMiR.

## 2. Method

### 2.1. DisiMiR

DisiMiR starts with the miRNA expression data of a specific disease and proceeds in four major steps. (1) A consensus-based network inference algorithm that integrates (1.1–1.5) five different network inference methods is used to infer the Consensus Influence Network from that expression data. This network is used to calculate (2.1) whole-network influence and (2.2) disease-specific influence, where (2.1) whole-network influence is a miRNA’s comparative influence relative to other miRNA’s, and (2.2) disease-specific influence is a disease-associated miRNA’s comparative influence relative to other disease-associated miRNAs. These two characteristics are aggregated with (3) miRNA conservation data and (4) mRNA target information to create a training dataset for (5) an AdaBoost model, which integrates whole-network influence (2.1), disease-specific influence (2.2), conservation (3), and mRNA targeting (4) to predict pathogenic miRNAs. 

### 2.2. DisiMiR Algorithm Outline

Consensus-based Network Inference clr [16]mrnet [17]mrnetb [18]aracne [19]GENIE3 [20]
Influence Inference Whole-network influenceDisease-specific influence
miRNA ConservationmRNA Target InformationCausal miRNA Prediction AdaBoost ModelPerformance EvaluationValidation of False Positive miRNAs using Recent Literature

#### 2.2.1. Consensus-Based Network Inference

DisiMiR starts with miRNA expression data from a particular disease of interest and is agnostic to the tissue of the dataset. In our present analysis, we apply DisiMiR on three different publicly available cancer datasets and one Alzheimer’s dataset from the Gene Expression Omnibus, but in principle, any similar dataset could be used. In the consensus-based network inference step, we use five different biological network inference algorithms on the chosen disease-specific miRNA expression data. We then combine the five different networks inferred using a normalized average (Figure 1). We will briefly describe the five different network inference algorithms employed below.

CLR [16], mrnet [17], mrnetb [18], and aracne [19] are information-theoretic network inference approaches that use the mutual information matrix to infer the regulatory network. We implement CLR, mrnet, mrnetb, and aracne with the minet Bioconductor package (v. 3.51.0) [21].

GENIE3 is a tree-based network inference approach that decomposes the network prediction problem over p nodes by creating p different regression problems [20]. For each node, GENIE3 trains a tree-based ensemble method to predict that node’s expression level based on the expression level of all other nodes in the network. The importance of a node in predicting the target node’s expression level indicates a putative regulatory link. These putative links are then aggregated over all nodes, and then ranked to construct the network. We implement GENIE3 with the GENIE3 Bioconductor package (v. 3.13) [20,22].

The output of each of these methods is an m × m adjacency matrix, where m is the number of miRNAs in the miRNA expression dataset, and an edge weight [i, j] represents the confidence that node i influences node j. By combining diverse network identification algorithms in a consensus-based approach, we increased the robustness of the inferred network.

Each of the five networks has edge weights scaled to [0, 1]. These are then averaged and pruned, retaining only edges with a weight of 0.7 or higher to obtain the Consensus Influence Network (1). We chose a relatively high cut-off threshold to retain only high-confidence edges, as is common practice in network inference algorithms [8].

#### 2.2.2. Influence Inference

We devised a miRNA *influence inference* algorithm I() that operates upon an influence network. With this influence inference algorithm, we calculate two types of influence: (2.1) whole-network influence and (2.2) disease-specific influence. The whole-network influence of a miRNA is obtained by computing I() on the entire Consensus Influence Network. To compute the disease-specific influence, we define the Disease-Specific Network as follows. The Disease-Specific Network is constructed from the Consensus Influence Network by pruning away all miRNAs and their incident edges not associated with the disease in the HMDD (Figure 2). The disease-specific influence of a miRNA is the obtained using I() on the Disease-Specific Network, which only contains disease-associated miRNAs and their edges. Disease-specific influence not only encodes whether a given miRNA is disease-associated, but also provides another dimension to look at a miRNA’s influence.

We then compute the whole-network influence of a given node in its network, using I() and the Consensus Influence Network, as follows. I() computes influence of a node as the weighted sum of the number of children of each node in its reachable network, where that sum is weighted by the length of the shortest path from the original node to a target node. We compute the disease-specific influence of a disease-associated miRNA similarly, using the Disease-Specific Network. We describe these computations in more detail in the following algorithm.
**Algorithm 1: Influence Inference I()**.
#For every miRNAfor m in V:  C_m_ = 0  #Add the immediate influence of all the miRNAs in a miRNA’s reachable network  for m’ in V’:    C_m_ + = n_m’_^(1/d)  end forend forV = all miRNAsm = a given miRNAV’ = all the miRNAs in the reachable network of mm’ = a given miRNA in the reachable network of mC_m_ = the influence of mn_m’_ = the number of children m’ hasd = (the length of the shortest path from m’ to m) + 1

#### 2.2.3. miRNA Conservation

We also use miRNA conservation to predict pathogenic miRNAs. The authors of MDCAP discovered that miRNAs that were causally implicated in more diseases were generally more conserved across species [12]. MiRNAs that are highly conserved are more important to key biological functions, and thus have a higher probability of causing disease when they are dysregulated. Of course, this conjecture is a statistical generalization, and not an inviolable rule: just because a miRNA is highly conserved does not mean it will cause a disease. As a measure of miRNA cross-species conservation, we used the sum of the miRNA family size as defined in miRbase (Release 22.1) and a Levenshtein distance-based *sequence similarity* metric S() [23]. miRNA family size counts the number of similar miRNAs within species and across species, and is a rough proxy for evolutionary conservation [23]. MiRNAs that have more family members across multiple species are more conserved because of their ubiquitous presence across multiple genomes [8]. Sequence similarity between miRNAs in a given family is a more direct measure of evolutionary conservation. Inspired by LE-MDCAP’s use of the Levenshtein distance [15], a function that returns the number of edits that are required to make two strings equivalent, we use a Levenshtein distance-based metric that represents how similar a given miRNA’s sequence is to all the other sequences in its family, as shown in Algorithm 2. By combining these two metrics in a single sum, we hope to gain a more accurate picture of miRNA conservation.
**Algorithm 2: Sequence Similarity S().**#For every miRNAfor m in V:  S_m_ = 0  #Add the immediate influence of all the miRNAs in a miRNA’s reachable network  for m’ in F:    S_m_ + = (length(m) − lev(m, m’))/length (m)  end forend forV = all miRNA sequencesm = a given miRNA sequenceF = the miRNA family of mm’ = a given miRNA sequence in FS_m_ = the similarity between m and all other sequences m’ in F lev(a, b) = returns the Levenshtein distance between String a and String blength (a) = returns the length of String a

#### 2.2.4. mRNA Target Information

The final piece of biological information we use to predict pathogenic miRNAs is the number of predicted mRNA targets a given miRNA has. It is intuitively true that miRNAs that regulate more mRNA targets should also exert greater influence in their regulatory network. For example, transfection of human cells with miR-124, which is preferentially expressed in the brain, causes the expression profile to shift towards that of the brain, showing its influence over the cellular state [24]. Since highly influential miRNAs can cause changes in the cellular state through signaling cascades, they are more likely to cause disease when disrupted. We take the number of predicted targets a given miRNA has from the TargetScan (Release 7.2) database [25].

#### 2.2.5. Causal miRNA Prediction

##### AdaBoost Model

We then aggregate the (2.1) whole-network influence, (2.2) disease-specific influence, (3) miRNA conservation, and (4) mRNA target information using an AdaBoost model (Figure 3). This will learn appropriate weights for each of these four features (whole-network influence, disease-specific influence, miRNA conservation, and mRNA target information). We split the miRNA feature matrix into a training and testing set, where one-third of the disease causal miRNAs according to HMDD are placed in the testing set, and the rest are used for training. We train an AdaBoost classifier with 1500 estimators on the training set, and validate it on the testing set, using the sklearn 1.0.2 AdaBoost python package [26].

##### Performance Evaluation

DisiMiR’s predictions are a probability from 0 to 1, indicating whether a given miRNA is pathogenic or not. We measure performance primarily with the Area Under the Receiver Operator Characteristic Curve (AUC) using DisiMiR’s average probability predictions across 100 random splits (Figure 4). In order to convert DisiMiR’s probability predictions into classes, we use a threshold defined as the point on the AUC that minimizes the sum of errors—false positives and false negatives—while still maintaining at least 10 false positives. We use this minimum false positive condition because the hypotheses generated from DisiMiR’s false positives could be useful for further investigation. Using this threshold, we also compute the confusion matrix and hypergeometric *p*-value. DisiMiR’s average predictions across 100 random training and testing splits are used to calculate the AUC, confusion matrix, *p*-values, and confidence intervals. We additionally report the average feature importance for the three features (whole-network influence, disease-specific influence, and miRNA conservation) across 100 random splits, which informs us about the importance of each feature for miRNA disease causality prediction.

##### Validation of False Positive miRNAs Using Recent Literature

MiRNAs falsely identified by DisiMiR to be disease causal according to HMDD (i.e., false positive errors) were investigated as follows. Using the Entrez PubMed API, a public application programming interface (API) connected to the database of papers in PubMed, we searched for publications using the combination of the hypothesis miRNA and the disease upon which it was hypothesized to have a causal effect, e.g., “mir-145 breast cancer”. This returned a number of papers, which were then analyzed manually. If the query returned no papers, that hypothesis was ignored and removed from further consideration. Relevant papers were then reviewed to determine if the manuscript was reporting research that demonstrated a causal effect of the miRNA. We use the same criteria as in the original HMDD paper [9]: causal effects were considered present if the manuscript was reporting an intervention in cellular or animal models that lead to characteristics indicative of pathology, not including interventions that enhanced treatment effects. These interventions included knockout experiments on the miRNA, knockout experiments on a regulator of the miRNA, or transfection of a miRNA mimic. In most cases, the abstracts were sufficient to make such a determination; however, in a small minority, the complete manuscript was considered. We report the proportion of false positive miRNAs with causal associations supported by disease-relevant literature to those with no support in disease-relevant literature.

## 3. Data

### 3.1. MiRNA Expression Datasets

On the basis that DisiMiR relies on multiple sources of biological information to infer disease causality, multiple datasets were used to develop and test DisiMiR. Four publicly available miRNA expression datasets from the Gene Expression Omnibus (GEO) [27] were used to develop and demonstrate DisiMiR (Table 1). All miRNA expression datasets were processed according to the proper guidelines and regulations. Each one of these datasets was used separately as an input to DisiMiR. These four datasets described breast cancer, Alzheimer’s disease, gastric cancer, and hepatocellular cancer. These data were used as is; no additional preprocessing was performed. All samples were disease state tissue; no control tissue was used.

The breast cancer dataset contained 32 human breast cancer, single hormone receptor-positive patient samples of different subtypes. Expression profiling was implemented with the NanoString nCounter Human v3 miRNA Expression Assay, and expression data had already been normalized according to the global mean of the counts of positive controls and all miRNA genes [28].

The Alzheimer’s disease dataset contained 197 serum Mild Cognitive Impairment (MCI) samples of different subtypes. MCI is a clinical precursor to Alzheimer’s disease (AD), and many MCI patients convert to AD, although others remain stable. Expression profiling was implemented with the 3D-Gene Human miRNA V21 spotted oligonucleotide array, and expression data had already been signal-filtered and normalized according to the ratio of the average signal of three internal control miRNAs [29].

The gastric cancer dataset contained 1423 serum gastric cancer samples. Expression profiling was implemented using the 3D-Gene Human miRNA V21 spotted oligonucleotide microarray, and expression data had already been normalized by signal filtering. The GENIE3 network inference algorithm was run on a random subset of 200 samples; the full dataset overwhelmed GENIE3, and caused it to run without terminating [30].

The hepatocellular cancer dataset contained seven samples of hepatocellular carcinoma tissue. Expression profiling was implemented with the Agilent Human miRNA Microarray, and expression data had already been quantile normalized using Gene Spring Software 11.0 [31].

### 3.2. HMDD

Known disease-associated and disease causal miRNAs were taken from the Human microRNA Disease Database (HMDD) v3.2, a manually curated database of miRNAs linked to disease [9]. Disease-association information was used as an input to calculate the disease-specific influence (step 2.2), and disease causality information was used to train and validate the method (steps 4.1 and 4.2). Since some diseases have multiple names in HMDD (for example, breast cancer can go by “Breast Neoplasms” or “Carcinoma, Breast”), we chose the name that yields the most confirmed causal miRNAs for each disease.

The HMDD 3.2 Causality Database was created through a manual literature review based on the following criteria: (1) the corresponding study must contain gain-of-function and/or loss-of-function experiments of the given miRNAs; (2) functional experiments must be conducted in a cell line and/or animal disease model; (3) studies that show that miRNAs could enhance drug effects, but have no contributions to diseases, are excluded [9]. If a study shows that there exists a causal relationship between a miRNA and disease, this relationship is recorded in the database; however, the absence of evidence does not conclusively mean a causal relationship does not exist, but rather, a causal relationship has not yet been found. Since miRNA expression analysis and sequencing technologies are still relatively nascent compared to other technologies and modalities, we expect the number of causal relationships found due to interventional experiments to increase in the coming years, including new causal relationships that were previously excluded from the HMDD 3.2 Causality Database.

## 4. Results

### 4.1. Inferred Networks

We trained and ran DisiMiR on four separate datasets from GEO: three cancer datasets (hepatocellular, breast, and gastric) and an Alzheimer’s dataset. Consensus Influence Networks for each disease are shown in Figure 5. The largest of the networks, the hepatocellular cancer network (Figure 5a), had 194,126 edges with two main subnetworks: the dense cluster visible in the top left, and the sparser subnetwork in the middle. The gastric cancer network had 3563 edges, with one main connected component (Figure 5b). The Alzheimer’s disease network had 4642 edges, with one main connected component (Figure 5c). The breast cancer network had 42,536 edges, with two main subnetworks (Figure 5d): the dense cluster visible in the top left, and the sparse, smaller subnetwork in the middle.

### 4.2. Validation

DisiMiR models were then tested for their ability to determine if miRNAs in the testing set were pathogenic or not. This was adjudicated according to HMDD. We report here results from averaging over 100 random train/test splits (Table 2, Figure 6). DisiMiR was most accurate on the hepatocellular cancer dataset, with an AUC of 0.957 (95% confidence interval (CI) 0.950, 0.965). Accuracy on gastric cancer and breast cancer was similar, with AUCs of 0.853 (95% CI 0.825, 0.882) and 0.853 (95% CI 0.791, 0.860), respectively. DisiMiR performed poorest on the Alzheimer’s disease data, with an AUC of 0.794 (95% CI 0.702, 0.886).

DisiMiR was also able to differentiate disease-associated miRNAs from disease causal miRNAs (Figure 6). HMDD was used to identify a subset of miRNAs that were associated with each of the four diseases. Surprisingly, DisiMiR differentiated best in the Alzheimer’s dataset, with an AUC of 0.708 (significantly better than random guessing; hypergeometric *p*-value 8.58 × 10^−4^). The rest of the diseases performed similarly: hepatocellular cancer, gastric cancer, and breast cancer had an AUC of 0.665 (*p*-value 4.61 × 10^−5^), 0.646 (*p*-value 3.42 × 10^−8^), and 0.612 (*p*-value 0.016). 

In step 4 of the DisiMiR algorithm, AdaBoost was used to learn a weighting to the four major types of information employed: (1) whole-network influence; (2) disease-specific influence; (3) miRNA conservation; and (4) target information. These weights, or feature importance (Table 2), reveal that each piece of biological information plays a non-trivial role in helping the model predict pathogenic miRNAs. miRNA conservation and target information seem to provide the most information, while the influence metrics provided the least information. 

DisiMiR learns disease-specific models during training. In Appendix A, we show the overlap of the miRNAs predicted to be causal for each of the four diseases, and compare this to the overlap between the causally associated miRNAs in HMDD. In general, DisiMiR predicted different miRNAs would be causal for each condition, with some overlap for the three cancer datasets. This effect is clearly seen in hepatocellular cancer: even though there are only 97 miRNAs causally associated with hepatocellular cancer in HMDD, DisiMiR predicts 241 unique miRNAs are causally associated with hepatocellular cancer. These figures, combined with the evidence that DisiMiR learns different feature importance for each disease, suggest that DisiMiR is, in fact, learning disease-specific models.

### 4.3. Hypothesis Generation with DisiMiR

Due to the fact that the HMDD database of pathogenic miRNAs is necessarily incomplete, we wanted to see whether the hypotheses that DisiMiR generated have been confirmed to be causal in recent literature. A miRNA was considered hypothesis-worthy if it was a false positive in the Confusion Matrix averaged over 100 random splits. Of the miRNAs that returned papers for a given disease, many of them were found to be causal. Out of all the false positives that had papers in PubMed, 78.4% of them were found to be causal (Table 3). This result shows that DisiMiR’s predictions can generalize outside of its training distribution, potentially identifying novel pathogenic miRNAs. 

## 5. Discussion

In this study, we present a method that flexibly and efficiently predicts pathogenic miRNAs from a disease context miRNA expression dataset. This method has possible diagnosis, research, and therapeutic applications. MiRNA drug targets are a promising new modality for drug discovery, yet there are few computational pipelines for miRNA drug target discovery. DisiMiR enables researchers to identify functional miRNAs in their own miRNA expression datasets for the elucidation of the underlying biological mechanisms, as well as to discover potential miRNA drug targets.

DisiMiR performed better on some disease datasets than it did on others. Somewhat surprisingly, the number of samples in each disease dataset did not seem to influence performance. DisiMiR was most accurate on the hepatocellular cancer dataset; the performance on the breast cancer and gastric cancer datasets was roughly the same, and the performance on the Alzheimer’s disease dataset was the worst. Two factors emerge as contributing explanations. First, how well a disease is characterized in the literature leads to more positive examples in the HMDD, which should, in turn, increase the method’s performance on that disease by providing more positive samples for the method to train on. This explanation is supported by the fact that hepatocellular cancer had the most positive samples in its testing set (395), followed by gastric cancer (270), breast cancer (151), and then Alzheimer’s disease (40). Additionally, we implemented DisiMiR on two other diseases, tuberculosis (10) and heart failure (3), and both of these disease datasets failed to train a statistically significant classifier. Second, miRNAs may play roles of differing importance in different diseases. At the biological level, miRNAs may be more involved in the progression of cancers, but less involved in the progression of neurological disorders such as Alzheimer’s disease. The role of miRNAs in cancer has been well established, and many miRNAs have been found to have a functional role in disease progression [32]. However, the role of miRNAs in Alzheimer’s disease is less characterized, and fewer miRNAs have been found to have functional significance [33]. With the decision threshold we used, DisiMiR skews towards specificity over sensitivity. This threshold resulted in high-quality false positive predictions, as 78.4% of our false positives were actually found to be positive according to the literature, but low overall sensitivity, especially compared to the AUC. For example, DisiMiR achieved a sensitivity of 0.238 on the breast cancer dataset. An AUC represents the spectrum of possible trade-offs a classifier can make between sensitivity and specificity, which may be tuned in various ways. By choosing a different threshold, we can achieve a sensitivity of 0.662 with little loss in specificity (Appendix A). The decision to make DisiMiR specificity- or sensitivity-favored is application dependent. For example, if a small set of false positive hypotheses are needed for further analysis, then specificity should be favored, but if a large set of false positive hypotheses are needed for a literature search, then sensitivity should be favored.

There were several alternate algorithmic design and parameter choices that DisiMiR could have implemented. We chose 0.7 as the threshold for the Consensus Influence Network edges over higher thresholds, such as 0.9, because higher thresholds resulted in networks that were too sparse. In the influence inference algorithm I(), we strictly used the topology of the consensus-based network, instead of taking the edge weights into account. Biological systems are incredibly complex, and there is no general way to accurately quantify the influence one regulatory element has on another. The authors of miRfluence attempted such an approach, yet they found that quantifying the edge weights yielded little improvement [7]. Additionally, we experimented with a hill-climbing procedure to optimize the weights used to average the consensus-based network, but DisiMiR’s performance improved negligibly. Finally, we tried different machine learning models, such as Logistic Regression, Random Forest, and an ensemble of weak AdaBoost classifiers, with different hyperparameters, and we found that a single AdaBoost classifier with 1500 estimators performed best on our datasets.

In conclusion, DisiMiR was able to successfully identify disease causal miRNA in four publicly available datasets, at rates higher than prior software. Furthermore, DisiMiR was also able to differentiate disease causal miRNAs from merely disease-associated miRNAs. DisiMiR can be used by researchers with miRNA expression data for a particular disease to identify disease causal miRNAs for further investigation. Future work could consist of exploring additional potential features to be used to predict pathogenicity, addressing DisiMiR’s low sensitivity, and investigating if certain network properties of the influence graphs impact the results. Utilizing biological insights, our work represents an important step forward in predicting disease causal miRNAs, and we believe DisiMiR will be a useful tool for future research into the regulatory mechanisms that drive disease.

## Figures and Tables

**Figure 1 ncrna-08-00045-f001:**
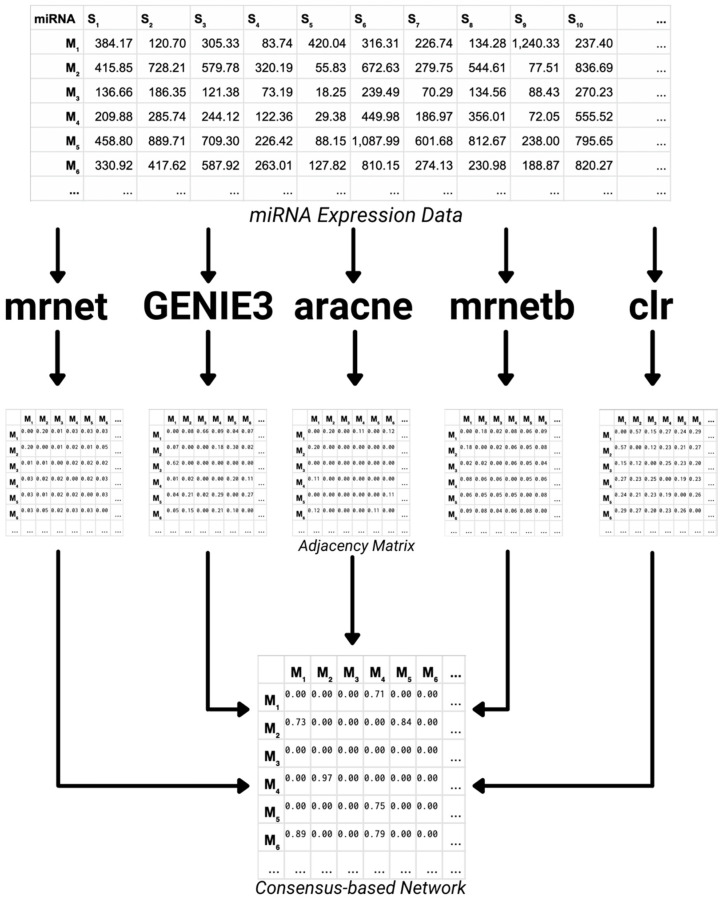
The consensus-based network inference algorithm computes the Consensus Influence Network from the average of five different networks inferred from the miRNA expression data. The GENIE3, mrnet, aracne, mrnetb, and clr network inference algorithms are used to infer five different networks, these are then aggregated into the Consensus Influence Network with a normalized average. The edges of the Consensus Influence Network are then pruned, retaining only edges with a weight higher than 0.7.

**Figure 2 ncrna-08-00045-f002:**
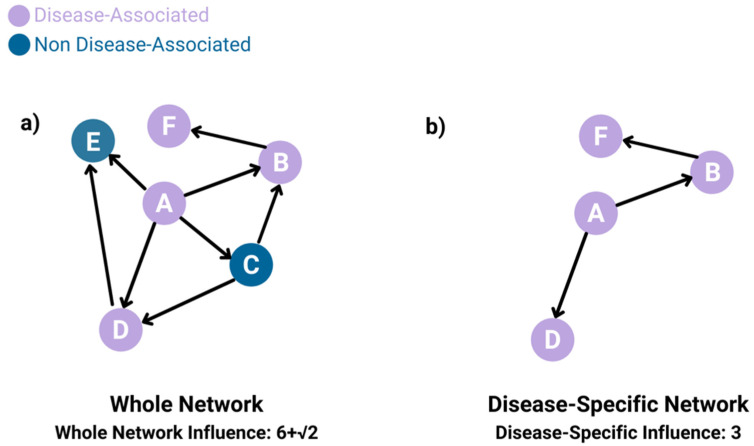
Whole-network influence uses I() on the Consensus Influence Network, while disease-specific influence uses I() on the Disease-Specific Network, with I() described in Algorithm 1. (**a**) Calculation of whole-network influence for node A. Node A has four children, which is raised to the power of 1. Node B and Node D each have one child, which is raised to the power of ½. Node C has two children, which is also raised to the power of ½. This gives a total whole-network influence of 6 + √2 for node A’s whole-network influence. (**b**) Calculation of disease-specific influence for node A. The Disease-Specific Network is created by removing all the nodes that are not disease-associated and their incident edges. Node A has two children, and Node B has one child. This gives a total disease-specific network example of 3.

**Figure 3 ncrna-08-00045-f003:**
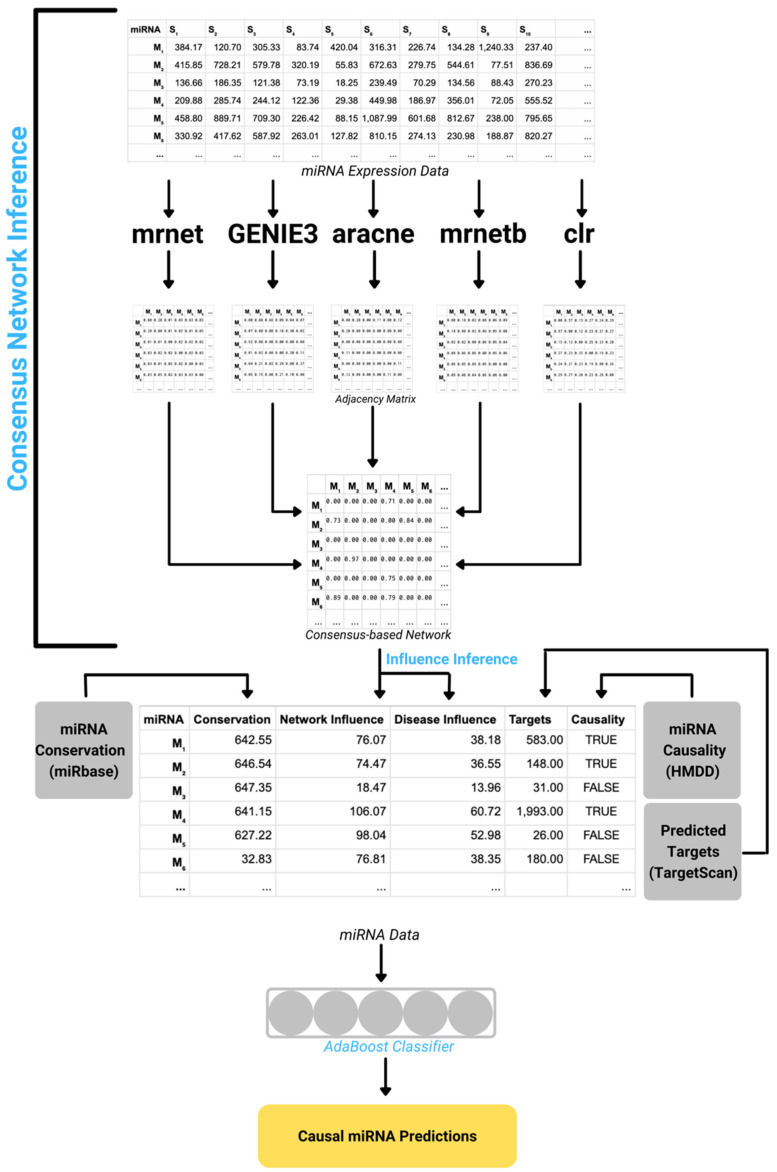
For a disease of interest, DisiMiR constructs a miRNA feature matrix, including whole-network influence, disease-specific influence, and miRNA conservation, in order to infer pathogenic miRNAs. The consensus-based network inference algorithm infers the Consensus Influence Network from the miRNA expression data. This network is used to calculate whole-network and disease-specific influence. These two characteristics are combined with miRNA conservation data from miRbase, target information from TargetScan, and miRNA causality information from HMDD, to create a training dataset. An AdaBoost model is then trained on this dataset to predict disease causal miRNAs.

**Figure 4 ncrna-08-00045-f004:**
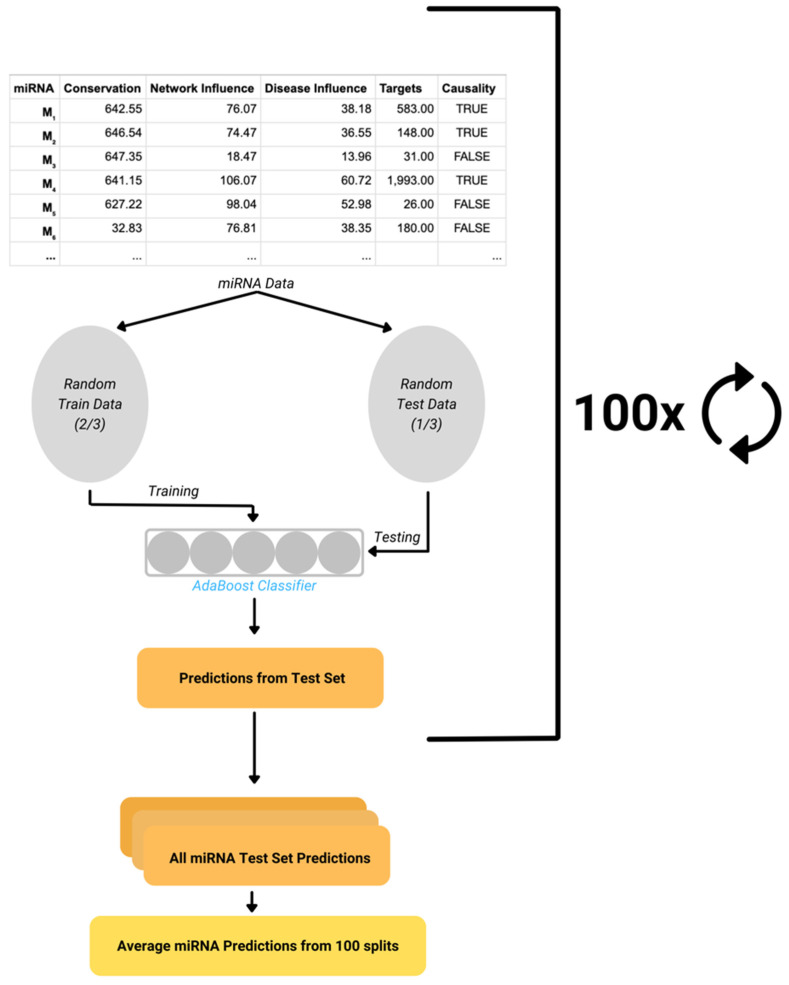
DisiMiR is run 100 times with different, random training and testing data splits. The predictions for each miRNA are averaged over 100 splits.

**Figure 5 ncrna-08-00045-f005:**
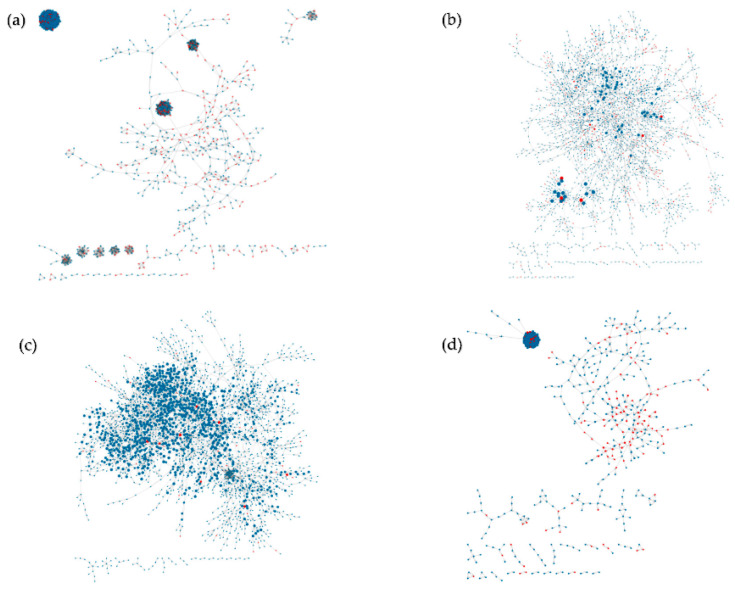
The consensus-based networks (step 1) were visualized in Cytoscape. Red-colored nodes are pathogenic; all other nodes are non-pathogenic. Node size is scaled by the node’s network influence. (**a**) Hepatocellular cancer, (**b**) gastric cancer, (**c**) Alzheimer’s disease, (**d**) breast cancer.

**Figure 6 ncrna-08-00045-f006:**
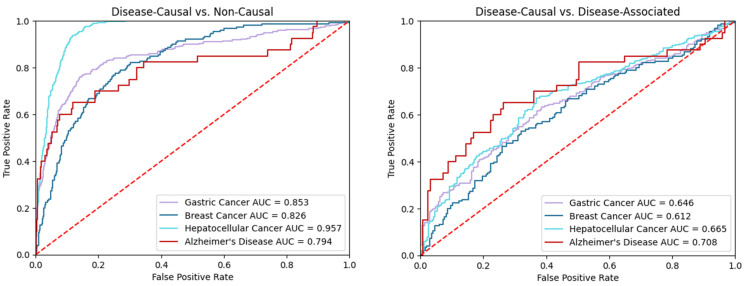
These AUCs demonstrate that DisiMiR can differentiate between disease causal and non-causal miRNAs, as well as disease causal and disease-associated miRNAs. **Left**: The AUCs from DisiMiR’s average predictions across 100 random splits. **Right**: The AUCs from DisiMiR’s average predictions across 100 random splits for disease-associated miRNAs only.

**Table 1 ncrna-08-00045-t001:** Disease Dataset Information.

HMDD Disease Name	Expression Profiling Method	Tissue Type	Number of Samples	Disease Causal	Disease Associated	Disease Irrelevant	Number of miRNAs	GEO Accession Number	Citation
Breast Neoplasms	NanoString nCounter Human v3 miRNA Expression Assay	Breast Cancer Tissue	32	151	348	480	828	GSE155362	Kunc, M. et al. [28]
Alzheimer’s Disease	3D-Gene Human miRNA V21 spotted oligonucleotide array	Serum	197	40	165	2356	2521	GSE150693	Shigemizu, D. et al. [29]
Gastric Neoplasms	3D-Gene Human miRNA V21 spotted oligonucleotide microarray	Serum	1423	270	478	2046	2524	GSE164174	Abe, S. et al. [30]
Carcinoma, Hepatocellular	Agilent Human miRNA Microarray	Liver Cancer Tissue	7	395	636	1933	2569	GSE108724	Zhu, H.-R. et al. [31]

**Table 2 ncrna-08-00045-t002:** Validation Metrics for each Disease Dataset across 100 random train/test splits.

Disease	AUC	Disease Causal Accuracy				Feature Importance			
		True Negative	False Positive	False Negative	True Positive	Disease Influence	Network Influence	miRNA Conservation	Number of Targets
Breast Cancer	0.826	647	30	115	36	0.121	0.197	0.386	0.297
Alzheimer’s Disease	0.794	2473	8	34	6	0.007	0.115	0.506	0.372
Hepatocellular Cancer	0.957	1999	175	71	324	0.183	0.197	0.379	0.241
Gastric Cancer	0.853	2236	18	200	70	0.040	0.187	0.371	0.403

**Table 3 ncrna-08-00045-t003:** Percentage of false positive miRNAs that were causal according to recent literature.

Disease	Number of Causal miRNAs	Number of Non-Causal miRNAs Mentioned in Literature without Causal Evidence	Percent Causal miRNAs	Number of miRNAs Unmentioned in Disease Literature	Total False Positives
Breast Cancer	24	5	82.8%	1	33
Gastric Cancer	14	4	77.8%	0	18
Hepatocellular Cancer	121	33	78.6%	21	155
Alzheimer’s Disease	1	2	33.3%	5	33

## Data Availability

All miRNA expression datasets used to produce the results are available in GEO, and their GEO Accession numbers are listed in Table 1.

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
