# Peer review of "DisiMiR: Predicting Pathogenic miRNAs Using Network Influence and miRNA Conservation"

_ncrna, 2022, doi:10.3390/ncrna8040045_

Round 1
Reviewer 1 Report
The authors describe a computational method to predict disease-causal miRNAs from disease-irrelevant miRNAs and disease-associated but not causal miRNAs. Biological features used for ML training included global or domestic network influences constructed from miRNA expression datasets, miRNA conservation derived from family member sizes, and a disease causality label recorded in HMDD database. Cross-validation and AUCs were reported, and for Alzheimer’s disease, the ensemble DisiMiR model was reported for a validation dataset. Overall it was an attractive experimental design and the application of the tool could be valuable for the miRNA-disease research field. However, several points are worth of discussion:
- In Introduction, the authors mentioned few methods existed for predicting miRNA causality. In fact, at least one such method existed, the LE-MDCAP, which was published in 2021.
- How many miRNAs were causal, merely associated, or disease irrelevant in each disease dataset? This doesn’t seem to be clear in the manuscript.
- The reviewer calculated the model’s sensitivity and specificity based on the numbers in Table 2, and found the sensitivity values were relatively low (0.23 for breast cancer, 0.3 for Alzheimer’s disease, and 0.47 for gastric cancer) except for hepatocellular cancer (0.86). For example, for breast cancer, there was supposed to be 151 causal miRNAs (True Positive + False Negative), and the model captured only 35 (True Positive). This could indicated that the features used as input for model construction was not sufficiently informing?
- For measuring conservation, why not directly use sequence similarity but rather sizes of the family members? While the latter is a proxy for evolutionary conservation, the former is a more accurate measure.
- miRNA is known to bind to the 3’ UTR of target mRNAs to induce mRNA degradation and therefore translational repression. In the authors’ method, the network influence of miRNA was solely measured by its impact on other miRNAs’ expression, and targets are not considered. The reviewer suspects that including the miRNA targets and weighting their importance in the diseases will improve the model performance, and this may shed light on how to better discern disease-causal from disease-associated miRNAs.
- The consensus network, as shown in Figure 1, how to explain that miRNA’s influence on itself was different for each miRNA (the numbers shown in diagonal) ? For example, in the table, m1’s influence on itself was 0.25, while for m2 it was 0.84.
- The consensus networks in Figure 5 largely show 2 interesting topologies, with (1) hepatocellular cancer and breast cancer having independent, dense miRNA clusters as well as a loosely connected net, and (2) gastric cancer and Alzheimer’s disease having more evenly connected miRNAs. This doesn’t seem to relate to the number of causal miRNAs, number of measured miRNAs, or sample sizes. Any biological explanations for the topology differences? What are the overlaps between the causal or associated miRNAs between diseases?
Author Response
Review #1.
The authors describe a computational method to predict disease-causal miRNAs from disease-irrelevant miRNAs and disease-associated but not causal miRNAs. Biological features used for ML training included global or domestic network influences constructed from miRNA expression datasets, miRNA conservation derived from family member sizes, and a disease causality label recorded in HMDD database. Cross-validation and AUCs were reported, and for Alzheimer’s disease, the ensemble DisiMiR model was reported for a validation dataset. Overall it was an attractive experimental design and the application of the tool could be valuable for the miRNA-disease research field. However, several points are worth of discussion:
- In Introduction, the authors mentioned few methods existed for predicting miRNA causality. In fact, at least one such method existed, the LE-MDCAP, which was published in 2021.
Response 1: We thank the Reviewer for bringing this relevant method to our attention. We’ve added a brief review of LE-MDCAP in our introduction.
- How many miRNAs were causal, merely associated, or disease irrelevant in each disease dataset? This doesn’t seem to be clear in the manuscript.
Response 2: We appreciate the Reviewer’s suggestion, and have added this information to Table 1.
- The reviewer calculated the model’s sensitivity and specificity based on the numbers in Table 2, and found the sensitivity values were relatively low (0.23 for breast cancer, 0.3 for Alzheimer’s disease, and 0.47 for gastric cancer) except for hepatocellular cancer (0.86). For example, for breast cancer, there was supposed to be 151 causal miRNAs (True Positive + False Negative), and the model captured only 35 (True Positive). This could indicate that the features used as input for model construction was not sufficiently informing?
Response 3: We agree with the Reviewer that our results are limited by the information latent in the input features, and DisiMiR does seem to have relatively low sensitivity values in general. However, DisiMiR has quite good AUCs, and by choosing a different decision threshold, we can increase the sensitivity values. Depending on the use-case, changing the threshold can make DisiMiR specificity-favored or sensitivity-favored, and, in the end, this decision is application dependent.
After adding the sequence similarity and target information to the pipeline (Reviewer points #4 and #5, below), and by choosing different thresholds, we did increase sensitivity with little losses in specificity. We have included this data in Supplementary Table 1. We have also added additional discussion of this issue in the Discussion.
- For measuring conservation, why not directly use sequence similarity but rather sizes of the family members? While the latter is a proxy for evolutionary conservation, the former is a more accurate measure.
Response 4: We agree with the Reviewer that sequence similarity may provide a more accurate measure of miRNA conservation. Based on the Reviewer’s suggestions, we implemented a new metric to measure evolutionary conservation. This metric is equal to the former metric, the size of the miRNA family, plus a Levenshtein-based sequence similarity metric (following the excellent reference for LE-MDCAP, as pointed out above). This new similarity metric is defined in section 2.2.3, Algorithm 2.
We used the Levenshtein distance function because we saw that it is the best function to compare the similarities of strings of different lengths from the LE-MDCAP paper. Normally, the Levenshtein distance is inversely proportional to the similarity of the two strings. Since we want the Levenshtein distance to be directly proportional to the similarity of the two strings, we then normalize the Levenshtein distance according to the length of the sequence of the miRNA of interest.
- miRNA is known to bind to the 3’ UTR of target mRNAs to induce mRNA degradation and therefore translational repression. In the authors’ method, the network influence of miRNA was solely measured by its impact on other miRNAs’ expression, and targets are not considered. The reviewer suspects that including the miRNA targets and weighting their importance in the diseases will improve the model performance, and this may shed light on how to better discern disease-causal from disease-associated miRNAs.
Response 5: We than the Reviewer for their excellent suggestion, and we agree with the Reviewer that general information on a miRNA’s targets could be important for its causal nature, and have included the number of predicted targets according to TargetScan as an additional informative feature.
Response 4 and 5: Adding these two new metrics to our pipeline, performance increases slightly on the disease-causal vs non-causal task (where AUCs increase significantly for Alzheimers and slightly for the rest of the diseases) and significantly on the disease-causal vs disease-associated task (where AUCs increase significantly for all three diseases except for Alzheimers where AUC slightly decreases). We have updated our results throughout, to reflect this improvement.
Both the conservation and target metrics have high feature importances across all diseases, as shown in Table 2, which suggest that they contain important information for disease-causal miRNA prediction. Additionally, the Levenshtein-distance based metric is a more sophisticated, accurate measure for evolutionary conservation, which merits its addition into the DisiMiR pipeline.
- The consensus network, as shown in Figure 1, how to explain that miRNA’s influence on itself was different for each miRNA (the numbers shown in diagonal) ? For example, in the table, m1’s influence on itself was 0.25, while for m2 it was 0.84.
Response 6: We apologize for the confusion, as we intended this image to be only for didactic purposes since it does not represent a real association between miRs and disease. To avoid future confusion, we have fixed the schematic to now show real numbers computed from the consensus network inference algorithm.
- The consensus networks in Figure 5 largely show 2 interesting topologies, with (1) hepatocellular cancer and breast cancer having independent, dense miRNA clusters as well as a loosely connected net, and (2) gastric cancer and Alzheimer’s disease having more evenly connected miRNAs. This doesn’t seem to relate to the number of causal miRNAs, number of measured miRNAs, or sample sizes. Any biological explanations for the topology differences? What are the overlaps between the causal or associated miRNAs between diseases?
Response 7: We thank the Reviewer for their insightful comment. We agree that the broad topological differences do not seem to correlate with DisiMiR’s performance on that disease dataset. It may be due to structural differences in the quality of the miRNA expression data or it may be due to the difference in the regulatory structures of the respective diseases. For example, the gastric cancer network also seems to have a dense cluster near the lower left corner of the image, though it is not as dense as the other cancer networks, while the Alzheimer’s disease network doesn’t have any dense clusters at all. We have added a (brief) speculation to the end of the Discussion where we note that network properties could be investigated in the future for their effect on performance.
Regarding the overlap, Venn diagrams showing the overlap between the miRNAs that DisiMiR predicts as causal for each disease and the overlap between the causally-associated miRNAs in HMDD for each disease are available in Supplementary Figure 1. We have also included a paragraph in the Results discussing the (small) overlap of the miRs predicted to be causal by DisiMiR.
Reviewer 2 Report
In this manuscript, Wang and McGeachie present a new software that is kind of sufficient to predict pathogenic miRNAs. This manuscript is ok in my opinion. The false-positive results of this software are still high, but I believe it should be a consequence that the causal relationships between miRNAs and diseases are not clear. Meanwhile, I think it's a nice try for this study area, although it would not cause a dramatic influence on that field or shed the light on future research. Honestly, I still think this paper is acceptable for this journal but I can not find any scientific hypotheses in the paper.
Some sentences aren't very academic at least not very professional. For example, in Line 47-50, the authors spend a whole sentence describing the benefit of the software, it's improper and not necessary. I strongly suggest a deep revision of the writing style.
Author Response
Review #2
In this manuscript, Wang and McGeachie present a new software that is kind of sufficient to predict pathogenic miRNAs. This manuscript is ok in my opinion. The false-positive results of this software are still high, but I believe it should be a consequence that the causal relationships between miRNAs and diseases are not clear. Meanwhile, I think it's a nice try for this study area, although it would not cause a dramatic influence on that field or shed the light on future research. Honestly, I still think this paper is acceptable for this journal but …
- I can not find any scientific hypotheses in the paper.
Response 1: We thank the reviewer for pointing out this omission! We have included a general hypothesis for our work in line 52 (Introduction). Additionally, DisiMiR has as its output a number of potential miRNAs that are hypothesized to have causal significance to the disease in question. The full list of these, for the four diseases treated here, are included in supplemental data files.
- Some sentences aren't very academic at least not very professional. For example, in Line 47-50, the authors spend a whole sentence describing the benefit of the software, it's improper and not necessary. I strongly suggest a deep revision of the writing style.
Response 2: We have rewritten the paper as suggested, making a number of changes we think bring the writing to a more scientific style, including Lines 47-50.